# Inverse Regulation of Lipocalin-2/24p3 Receptor/SLC22A17 and Lipocalin-2 Expression by Tonicity, NFAT5/TonEBP and Arginine Vasopressin in Mouse Cortical Collecting Duct Cells mCCD(cl.1): Implications for Osmotolerance

**DOI:** 10.3390/ijms20215398

**Published:** 2019-10-30

**Authors:** Stephanie Probst, Bettina Scharner, Ruairi McErlean, Wing-Kee Lee, Frank Thévenod

**Affiliations:** 1Department of Physiology, Pathophysiology & Toxicology and ZBAF (Centre for Biomedical Education and Research), Faculty of Health, School of Medicine, Witten/Herdecke University, Stockumer Str 12 (Thyssenhaus), D-58453 Witten, Germany; Stephanie.Probst@uni-wh.de (S.P.); Bettina.Scharner@uni-wh.de (B.S.); ruairi.mcerlean@student.manchester.ac.uk (R.M.); Wing-Kee.Lee@uni-wh.de (W.-K.L.); 2Faculty of Biology, Medicine and Health, School of Biological Sciences, University of Manchester, Oxford Rd, Manchester M13 9PL, UK

**Keywords:** kidney, hypertonicity, osmotic stress, lipocalin-2, lipocalin-2 receptor, lipopolysaccharide, TonEBP, CREB

## Abstract

The rodent collecting duct (CD) expresses a 24p3/NGAL/lipocalin-2 (LCN2) receptor (SLC22A17) apically, possibly to mediate high-affinity reabsorption of filtered proteins by endocytosis, although its functions remain uncertain. Recently, we showed that hyperosmolarity/-tonicity upregulates SLC22A17 in cultured mouse inner-medullary CD cells, whereas activation of toll-like receptor 4 (TLR4), via bacterial lipopolysaccharides (LPS), downregulates SLC22A17. This is similar to the upregulation of *Aqp2* by hyperosmolarity/-tonicity and arginine vasopressin (AVP), and downregulation by TLR4 signaling, which occur via the transcription factors NFAT5 (TonEBP or OREBP), cAMP-responsive element binding protein (CREB), and nuclear factor-kappa B, respectively. The aim of the study was to determine the effects of osmolarity/tonicity and AVP, and their associated signaling pathways, on the expression of SLC22A17 and its ligand, LCN2, in the mouse (m) cortical collecting duct cell line mCCD(cl.1). Normosmolarity/-tonicity corresponded to 300 mosmol/L, whereas the addition of 50–100 mmol/L NaCl for up to 72 h induced hyperosmolarity/-tonicity (400–500 mosmol/L). RT-PCR, qPCR, immunoblotting and immunofluorescence microscopy detected *Slc22a17*/SLC22A17 and *Lcn2*/LCN2 expression. RNAi silenced *Nfat5*, and the pharmacological agent 666-15 blocked CREB. Activation of TLR4 was induced with LPS. Similar to *Aqp2*, hyperosmotic/-tonic media and AVP upregulated *Slc22a17*/SLC22A17, via activation of NFAT5 and CREB, respectively, and LPS/TLR4 signaling downregulated *Slc22a17*/SLC22A17. Conversely, though NFAT5 mediated the hyperosmolarity/-tonicity induced downregulation of *Lcn2*/LCN2 expression, AVP reduced *Lcn2*/LCN2 expression and predominantly apical LCN2 secretion, evoked by LPS, through a posttranslational mode of action that was independent of CREB signaling. In conclusion, the hyperosmotic/-tonic upregulation of SLC22A17 in mCCD(cl.1) cells, via NFAT5, and by AVP, via CREB, suggests that SLC22A17 contributes to adaptive osmotolerance, whereas LCN2 downregulation could counteract increased proliferation and permanent damage of osmotically stressed cells.

## 1. Introduction

The chief site of urine concentration is in the collecting duct (CD) system [1]. There, the major effector in the regulation of renal water excretion is the antidiuretic hormone arginine vasopressin (AVP), which binds to the AVP type-2 receptor (V2R) and signals through cAMP [2]. AVP facilitates urinary concentration in the medullary and cortical CD of the kidneys, by increasing CD water permeability through the enhancement of aquaporin-2 (AQP2) water channel incorporation in the luminal membrane of principal cells, permitting water to flow passively along the osmotic gradient, from the tubule lumen to the interstitium [3]. AVP stimulation activates adenylyl cyclase, which results in increased cytosolic cAMP concentration and subsequent protein kinase A (PKA) activation. This, in turn, triggers the trafficking of intracellular storage vesicles expressing AQP2 to the luminal plasma membrane, within a range of 10–30 min (short-term regulation), and also increases *Aqp2* gene transcription, via the increased activity of cAMP-responsive element binding protein (CREB) and AP-1 [4,5,6], over a time period ranging from hours to days (long-term regulation).

For AVP to exert its effects on water transport in the CD, axial cortico–papillary osmotic gradients need to be generated through the accumulation of high interstitial concentrations of NaCl (300–400 mmol/L) and urea (> 600 mmol/L) [7,8]. Na^+^ reabsorption in the thick ascending limb results in a renal cortico–papillary osmotic gradient. However, this gradient exposes renal cells to substantial osmotic stress by causing numerous perturbations (reviewed in [9]). Cells can respond to high osmotic stress by activating adaptive mechanisms through various pathways that activate the transcription factor NFAT5 (also known as tonicity-named responsive enhancer binding protein (TonEBP or OREBP)), culminating in the accumulation of organic osmolytes and increased expression of heat shock proteins (reviewed in [9]). In addition to AVP, extracellular tonicity is pivotal in determining AQP2 abundance, through the activation of NFAT5, which boosts AVP-induced transcriptional activation of *Aqp2*. Conversely, activation of the NF-κB transcriptional factor by pro-inflammatory signals reduces *Aqp2* gene transcription (reviewed in [3,10]).

The CD is a site of ascending urinary tract infections (UTI). Lipocalin-2 (LCN2; also NGAL [human] or siderocalin/24p3 [rodent]) binds Fe^3+^ through association with bacterial siderophores, hence it plays an important role in innate antibacterial immunity [11]. Activation of the Toll-like receptor 4 (TLR4) on CD cells, by the bacterial wall component lipopolysaccharide (LPS), has been shown to induce LCN2 secretion to combat urinary bacterial infections [12]. A receptor for LCN2 (LCN2-R/SLC22A17/24p3-R) has been cloned (MM ~60kDa) [13], and is expressed in the apical membrane of distal convoluted tubules and CD [14]. Experimental evidence in cultured cells and in vivo [14,15] indicates that SLC22A17 is a high-affinity receptor, involved in protein endocytosis in the distal nephron [16]. In fact, the affinity of SLC22A17 to filtered proteins, such as LCN2 or metallothionein, is ~1000x higher than that of megalin [14] (reviewed in [16,17]), the high-capacity receptor for endocytic reabsorption of filtered proteins in the proximal tubule [18].

Our understanding of the physiological regulation of SLC22A17 and LCN2 expression in vivo is poor. Recent data, obtained by deep sequencing in micro-dissected nephrons, showed the highest SLC22A17 expression levels in the rat inner medullary CD (IMCD) compared to other nephron segments, whereas LCN2 levels were negligible [19]. Abundant localization of SLC22A17 in the CD [14] strongly implies a relationship with the hypertonic environment, and possibly regulation by AVP. Our recent data in a mouse IMCD cell line (mIMCD_3_) evidenced *Slc22a17*/SLC22A17 upregulation and *Lcn2*/LCN2 downregulation, induced by hyperosmolarity/-tonicity, suggesting adaptive osmotolerant survival, whereas *Lcn2*/LCN2 upregulation and *Slc22a17*/SLC22A17 downregulation, via TLR4, indicated protection against bacterial infections [20].

In the present study, the role of increased osmolarity/tonicity on SLC22A17and LCN2 expression was investigated in the mouse cortical CD (CCD) cell line mCCD(cl.1). The data indicate that regulation of SLC22A17 expression is analogous to AQP2, i.e., increased osmolarity/tonicity and AVP induced expression of SLC22A17, via activation of the transcription factors NFAT5 and CREB, respectively, which is counteracted by LPS/TLR4 signaling. Whereas NFAT5 mediates LCN2 downregulation elicited by hyperosmolarity/-tonicity, AVP reduces LCN2 expression and secretion evoked by LPS, through a posttranslational mode of action. The parallel regulation of AQP2 and SLC22A17 expression in mCCD(cl.1) cells suggest the role of SLC22A17 in vivo in urine concentration and/or osmotic stress adaptation.

## 2. Results

### 2.1. Hyperosmolarity Increases the Expression of Slc22a17/SLC22A17 in mCCD(cl.1) Cells

The CD can be separated into the CCD, outer medullary CD (OMCD) and IMCD. Each section of the CD contributes to urine concentration and is also exposed to varying degrees of osmotic stress. As a matter of fact, extracellular osmolarity increases from approximately 300 mosmol/L in the renal cortex, to, maximally, 1200 mosmol/L in the renal medulla. We have previously shown that increased extracellular osmolarity augments abundance of the LCN2 receptor SLC22A17 in a mouse IMCD cell line (mIMCD_3_) [20]. SLC22A17 is expressed apically along the rodent CD [14] as well as in the mCCD(cl.1) cell line [21]. To assess whether the CCD also harbors a similar positive regulatory relationship between extracellular osmolarity and SLC22A17 expression, mCCD(cl.1) cells were exposed to hyperosmotic medium and *Slc22a17*expression was determined. Indeed, increased osmolarity to 400 mosmol/L for 6 h (data not shown) and 12 h induced *Slc22a17* mRNA, as demonstrated by RT-PCR (Figure 1A) and qPCR (Figure 1B). Moreover, hyperosmolarity for 48 h increased plasma membrane expression of SLC22A17 protein (Figure 1C). This was associated with increased protein expression of SLC22A17 in microsomes of mCCD(cl.1) cells, that are enriched via the plasma membrane-located Na^+^/K^+^-ATPase (Figure 1D). In addition, Na^+^/K^+^-ATPase was also upregulated in cells exposed to hyperosmotic media, which indicates that an adaptive osmoprotective response to hyperosmolarity has been engaged [9]. In contrast, the ligand of SLC22A17, *Lcn2*, was downregulated by hyperosmolarity by RT-PCR (Figure 1E) and qPCR (Figure 1F), which is in line with recent data in mIMCD_3_ cells [20]. Overall, these data indicate that hyperosmolarity regulates *Slc22a17*/SLC22A17 expression in the CCD cell line mCCD(cl.1), whereas the expression of its natural ligand *Lcn2* is reduced, recapitulating the observations made in IMCD cells [20].

### 2.2. Hypertonicity Dependent Upregulation of Slc22a17/SLC22A17 and Downregulation of Lcn2/LCN2 are Mediated by NFAT5 in mCCD(cl.1) Cells

Increased extracellular osmolarity has previously been shown to increase abundance of the water channel *Aqp2 in vivo* [22,23], and in the mouse renal CD principal cell line mpkCCDcl4 (reviewed in [3,10]), and depends on increased nuclear activity of the transcription factor NFAT5 (TonEBP/OREBP) [24], which is also affected by osmolarity in vivo [25]. These findings were confirmed in the mCCD(cl.1) cell line: hyperosmolarity of 400 mosmol/L for 24 h upregulated *Aqp2* mRNA by RT-PCR (Figure 2A), and qPCR (Figure 2B), slightly increased NFAT5 (Figure 2C), suggesting the slow induction of NFAT5 protein [26,27], and hyperosmolarity of 400–500 mosmol/L induced translocation of cytosolic NFAT5 to nuclei (Figure 2D). To investigate whether the upregulation of *Aqp2* by hyperosmolarity is also mediated by NFAT5 in mCCD(cl.1) cells, RNAi was performed, using siRNAs against *Nfat5*, which efficiently abolished the hyperosmolarity-induced increase in both *Nfat5* mRNA (Figure 2E) and NFAT5 protein (Figure 2C) levels. Accordingly, as expected, *Nfat5* silencing almost abolished the induction of *Aqp2* mRNA elicited by hyperosmolarity (Figure 2F).

Since the central response pathway to changes in tonicity is NFAT5, we hypothesized that SLC22A17 and its ligand, LCN2, could be regulated by NFAT5 when extracellular osmolarity rises. Strikingly, induction of *Slc22a17* mRNA by hyperosmolarity was significantly reduced by *Nfat5* silencing after 6–12 h (Figure 3A), and expression of the SLC22A17 protein at the surface of mCCD(cl.1) cells was significantly decreased after 24 h hyperosmolarity (Figure 3B). In contrast, *Nfat5* silencing reversed the downregulation of *Lcn2* mRNA expression induced by hyperosmolarity. Figure 3C shows that hyperosmolarity induced an early decrease of *Lcn2* mRNA at 6 and 12 h, which was significantly reversed by *Nfat5* silencing at 6 h, and confirmed at the protein level by immunoblotting of LCN2 (Figure 3D). Hence, the transcription factor NFAT5, which regulates the expression of genes involved in osmotic stress, is largely responsible for up- and downregulation of *Slc22a17*/SLC22A17 and *Lcn2*/LCN2, respectively.

### 2.3. ddAVP Upregulates Slc22a17/SLC22A17 via cAMP/CREB Signaling in mCCD(cl.1) Cells

In addition to hyperosmolarity, the V2R also increases AQP2 protein abundance, to facilitate urinary concentration in the CCD and MCD [28,29], through a cAMP-responsive element (CRE) located in the *Aqp2* promoter, leading to AVP-induced *Aqp2* transcription [4,5,6]. This mechanism has been reported in mpkCCDcl4 cells [30,31]. Analogously, when mCCD(cl.1) cells were exposed to the specific V2R agonist ddAVP (10 nM) for 24 h in an isotonic medium, *Aqp2* mRNA increased 5.85 ± 0.71-fold, and this effect was significantly reduced to 2.60 ± 0.29 (means ± SEM of 7–11 experiments) by the potent inhibitor of CREB-mediated gene transcription, 666-15 (100–250 nM) [32] (Figure 4A). Notably, ddAVP slightly, but significantly upregulated *Slc22a17* mRNA (1.38 ± 0.06-fold; *n* = 7–11) in a 666-15 sensitive-manner (Figure 4A). The latter results were further confirmed at the protein level by quantifying surface expression of SLC22A17 protein in mCCD(cl.1) cells, treated exactly as described for the qPCR data shown in Figure 4A. Figure 4B shows that ddAVP increased SLC22A17 surface expression about two-fold, and the role of CREB activation was demonstrated by a significant reduction in SLC22A17 surface expression, by preincubation with 666-15 (Figure 4B). Consequently, gene regulatory mechanisms for SLC22A17, via the transcription factors NFAT5 and CREB, in response to osmotic stress, parallels that of *Aqp2*.

### 2.4. LPS Downregulates Slc22a17/SLC22A17 and Upregulates Lcn2/LCN2 in mCCD(cl.1) Cells

The renal CD is susceptible to bacterial infections that can ascend from the lower urinary tract during UTIs. The activation of TLR4 [33] in collecting ducts [34], mediated by NF-κB-mediated signaling [35], via bacterial LPS, has been shown in our earlier work to induce LCN2 upregulation and SLC22A17 downregulation in mIMCD_3_ cells, and suggested to protect IMCD cells against bacterial infections, and prevent autocrine death induction by LCN2 [20]. Here we investigated whether LPS also inhibits SLC22A17 expression in the CCD. When mCCD(cl.1) cells incubated in isotonic medium were exposed to LPS (5 μg/mL) for 12 h, *Slc22a17* mRNA was significantly reduced (Figure 5A), as measured by RT-PCR. Similarly, LPS reduced cellular and surface expression of SLC22A17, as well (Figure 5B). LPS at 100 ng/mL for 12 h yielded similar results.

Further corroborating our previous findings in IMCD cells, LPS (5 μg/mL for 12 h) significantly stimulated *Lcn2* mRNA expression in mCCD(cl.1) cells (Figure 5C), and increased LCN2 protein expression, when cells were exposed for 18 h to 5 μg/mL LPS (Figure 5D,E). Interestingly, LPS exposure time ≤ 12 h marginally increased LCN2 secretion. This high LPS concentration of 5 μg/mL is saturating, with regard to LCN2 secretion in mCCD(cl.1) cells (Appendix A). It has previously been shown to activate NF-kB dependent TLR4 signaling in cultured mouse CD cell lines, which involves rapid IκB-α degradation [36,37] (Appendix A), as well as in primary CD cells [38] (although additional activation of the non-canonical, TLR4-independent inflammasome pathway cannot be excluded). Secretion of LCN2 occurred predominantly apically, when LPS was applied to both the apical and basolateral compartments of confluent monolayers of mCCD(cl.1) cells grown on transwell filters (Figure 5E). Yet, unstimulated secretion was also predominantly apical (Figure 5E), confirming that constitutive, as well as regulated, LCN2 secretion largely targets the apical plasma membrane, as previously suggested [12].

### 2.5. ddAVP Posttranslationally Downregulates Unstimulated and LPS-Stimulated LCN2 in mCCD(cl.1) Cells

As shown in Figure 6A, LPS (5 μg/mL for 18 h) significantly reduced unstimulated and ddAVP-induced *Aqp2* mRNA expression in mCCD(cl.1) cells, thus confirming previous studies in mpkCCDcl4 cells [35]. Despite LPS stimulation of *Lcn2* mRNA levels (Figure 6B), ddAVP (10 nM) had no effect on *Lcn2* mRNA as determined by RT-PCR and confirmed by qPCR (Figure 6C). In addition, the CREB inhibitor 666-15 was ineffective (Figure 6C). Quite surprisingly, and in contrast to the mRNA data, both unstimulated and LPS-stimulated LCN2 protein expression and secretion were reduced by ddAVP (Figure 6D), suggesting a posttranslational effect of ddAVP on LCN2 expression and secretion. Indeed, co-application of cycloheximide (0.01 μg/mL for 3–6 h), a blocker of translational elongation [39], prevented the downregulation of LCN2 protein expression induced by ddAVP (Figure 6E), which indicates that ddAVP promotes LCN2 degradation following protein translation.

## 3. Discussion

It is interesting to note that SLC22A17 regulation is analogous to that of AQP2 in vivo and mpkCCDcl4 cells (reviewed in [3,10]). AVP stimulation of PKA activity increases abundance of *Aqp2*. PKA, in turn, activates *Aqp2* gene transcription, in part via increased CREB binding to cis elements of the *Aqp2* promoter. After longer periods of hypertonic challenge, NFAT5 also participates in increasing *Aqp2* gene transcription. In contrast, bacterial infection has been shown to interfere with *Aqp2* mRNA and AQP2 protein expression, via activation of the inflammatory pathway, involving NF-κB in vivo [40], and using LPS in cultured renal CCD mpkCCDcl4 [35]. Similarly, in mCCD(cl.1) cells, hyperosmolarity (Figure 1) and ddAVP (Figure 4A,B), respectively, increased SLC22A17 in an NFAT5- (Figure 3A,B) and CREB-dependent manner (Figure 4A,B), whereas LPS also reduced SLC22A17 expression (Figure 5A,B). The latter observation is similar to that of our previous study with mIMCD_3_ cells, where LPS reduced SLC22A17 expression as well [20], most likely via NF-κB activation [36].

In our previous study, mIMCD_3_ cells activated Wnt/TCF1/β-catenin signaling with inhibitory phosphorylation of GSK-3β, which caused SLC22A17 upregulation, in response to hyperosmolarity [20]. In silico analysis [41,42] indicated a putative TCF1 binding site 111 bases upstream of the *Slc22a17* gene, supporting the hypothesis that SLC22A17 is a downstream target of Wnt/β-catenin signaling. Is Wnt signaling linked to the central osmotic response pathway involving NFAT5? A study in HEK293 and mIMCD_3_ cells indicated that inhibitory phosphorylation of GSK-3β contributes to high NaCl-induced activation of NFAT5, and suggested upstream regulation of NFAT5 by Wnt/GSK-3β signaling [43]. Supporting this, a putative *Nfat5* DNA consensus motif (osmotic response element (ORE), or tonicity-responsive enhancer (TonE)) [44], was identified in the human *SLC22A17* promoter sequence ENSR00001455985/14:23820951-23822367 (^828^AGGAAAATGCCA^839^), which is compatible with the data obtained in Figure 3A,B. It would be worthwhile to test this putative ORE in future experiments, by site-directed mutagenesis of the sequence in a *Slc22a17* luciferase-reporter gene assay. Moreover, a search of the *Slc22a17* promoter sequence for CREB binding sites, with the tools JASPAR and PROMO [41,45], up to 2.7 kb upstream of the start codon, yielded a putative binding site at position -736 (Ralf Zarbock and Frank Thévenod, unpublished), supporting the results obtained in Figure 4.

The human *LCN2* gene (Ensembl ENSG00000148346; GenBank: X99133.1; 5869 bp) has ten putative regulatory promoter sequences. Similarly, as described for *Slc22a17*, we have also identified a putative *NFAT5* DNA binding sequence in a region of the human *LCN2* gene (^841^TGGAAAAAGGCT^852^), which could explain the *Lcn2* downregulation by hyperosmolarity in Figure 3C,D. In contrast, ddAVP did not influence *Lcn2* gene expression (Figure 6B,C). Rather, ddAVP reduced LCN2 protein expression by a posttranslational mechanism (Figure 6D,E).

It is interesting to note that NFAT5 regulates LCN2 and its receptor SLC22A17 in an inverse manner, namely by causing upregulation of the receptor and downregulation of the ligand. In mIMCD_3_ cells, hyperosmolarity induced an identical inverse regulation, via Wnt/β-catenin signaling [20], possibly because NFAT5 is downstream of Wnt/GSK-3β signaling [43] (see above). In addition, the oncogene BCR-ABL activated the JAK/STAT pathway in murine myeloblast-like cells, which increased LCN2, and repressed SLC22A17 expression in a Ras and Runx1-dependent manner [13,46]. Hence, several signaling pathways exist to inversely control LCN2 and its receptor, with different putative biological significance and outcomes (see below and [20] for a discussion).

Unexpectedly, the inverse regulation of *Slc22a17* and *Lcn2* at the gene level did not extend to ddAVP-mediated CREB regulation. Whereas CREB activation increased *Slc22a17*, ddAVP seemed to control LCN2 via a posttranslational mechanism. For *Aqp2*, AVP not only regulates its expression at the gene level via CREB, but also PKA-dependent phosphorylation of the protein triggers its increased proteasomal and lysosomal degradation in a negative feedback loop [47]. This is unlikely for LCN2, because PKA does not phosphorylate the protein in vitro [48].

What could be the physiological significance of SLC22A17 upregulation in the context of hyperosmotic stress and AVP-mediated urinary concentration? SLC22A17 mediates receptor-mediated protein endocytosis in the distal nephron [14]. Endocytosed proteins are trafficked to lysosomes [14], where they are degraded. We speculate that SLC22A17 promotes osmotolerance by feeding cells with amino acids as precursors/osmolytes, to maintain iso-osmolarity with the interstitium. However, this will require experimental proof. Downregulation of LCN2 in the same context would also foster cellular protection during hyperosmotic stress. It has been proposed that LCN2 upholds epithelial growth and proliferation (reviewed in [49]), which is associated with increased DNA replication and transcription. Osmotic stress induces DNA strand breaks, inhibits DNA repair and increases oxidative stress, which may all lead to increased mutation rates in dividing cells, unless proliferation is inhibited, allowing for efficient repair.

## 4. Materials and Methods

### 4.1. Materials

Lipopolysaccharides (LPS) from Escherichia coli (cat. # L3129), [deamino-Cys1, D-Arg8]-vasopressin acetate salt hydrate (ddAVP, desmopressin) cat. # V1005) and protease inhibitor cocktail (cat. # P8340) were obtained from Sigma–Aldrich (Taufkirchen, Germany). The 3-(3-Aminopropoxy)-N-[2-[[3-[[(4-chloro-2-hydroxyphenyl) amino]carbonyl]-2-naphthalenyl] oxy] ethyl]-2-naphthalenecarboxamide hydrochloride (666-15) (cat. # 5661) was obtained from Tocris Bioscience. Cycloheximide (cat. # ALX-380-269-G001) was obtained from Enzo Life Sciences (Lörrach, Germany). All other reagents were of the highest purity grade possible. Materials were dissolved in water, ethanol, or dimethyl sulfoxide (DMSO). In control experiments, solvents were added to cells at concentrations not exceeding 0.2%. Antibodies are listed in Table 1.

### 4.2. Methods

#### 4.2.1. Culture of mCCD(cl.1) Cells

The mouse (m)CCD(cl.1) cell line [50] was obtained from Dr. Edith Hummler (University of Lausanne, CH). Cells (passage 25–34) were cultured in Dulbecco’s modified Eagle’s medium (DMEM)/nutrient mixture F-12 (1:1) (Gibco cat. # 31330), supplemented with 2.5 mM l-glutamine, 15 mM Hepes, 5% fetal bovine serum (FBS), 100 U/mL penicillin, and 100 μg/mL streptomycin, 0.9 μM insulin (Sigma–Aldrich; cat. # I1882), 5 μg/mL apo-transferrin (Sigma–Aldrich; cat. # T2252), 10 ng/mL EGF (Sigma–Aldrich; cat. # E9644), 1 nM T3 (Sigma–Aldrich; cat. # T6397), and 50 nM dexamethasone (Sigma–Aldrich; cat. # D4902) [51]. Cells were cultured in 25 cm^2^ standard tissue culture flasks (Sarstedt, Nümbrecht, Germany) at 37 °C in a humidified incubator with 5% CO_2_, and passaging was performed twice a week upon reaching 90% confluence. Inhibitors (666-15, cycloheximide) were preincubated for 60 min.

#### 4.2.2. Osmolarity/Tonicity Experiments

Osmolarity is the measure of solute concentration per unit volume of solvent. Tonicity is the measure of the osmotic pressure gradient between two solutions across biological membranes. Unlike osmolarity, tonicity is only influenced by solutes that cannot cross this semipermeable membrane, because they create and influence the osmotic pressure gradient between the intracellular and extracellular compartments. Unless otherwise indicated, in osmolarity/tonicity experiments, cell lines were cultured for 24 h in a standard culture medium (=300 mosmol/L) after seeding. Then the medium was replaced with either normosmotic standard culture medium or hyperosmotic medium of 400–500 mosmol/L (by addition of 50 mmol/L NaCl from 3 M stock solutions), and cultured for up to 72 h (CCD are physiologically exposed to comparable hyperosmotic interstitial concentrations of NaCl [7,8]). NaCl initially does not cross the cell membrane, and therefore exerts an osmotic pressure gradient on the cells (hypertonicity). For simplification, the term “osmolarity” henceforth refers to osmolarity and tonicity.

#### 4.2.3. Transient Transfection

Cells were transiently transfected with *Nfat5* siRNA (sense primers for Stealth siRNA (Invitrogen, San Diego, CA, USA) 5′-GGUGUUGCAGGUAUUUGUGGGCAAU-3′, 5′-GGAUUCUAUCAGGCCUGUAGAGUAA-3′, 5′-CCUAGUUCUCAAGAUCAGCAAGUAA-3′ [24]) or siRNA duplex negative control (Eurogentec cat. # SR-CL000-005). For lipid-based transfections, 1.0 × 10^5^ mCCD(cl.1) cells were seeded in 6-well plates and transfected 24 h later at ~40% confluence with 5 nM siRNA, using Lipofectamine RNAiMAX (Thermo Fisher Scientific, Schwerte, Germany), according to the manufacturer’s instructions. After 24 h, the medium was exchanged to 300 or 400 mosmol/L media.

Transfection by electroporation was performed as described previously [52], with slight modifications. In brief, 2.0 × 10^6^ mCCD(cl.1) cells were electroporated at 300 mV and 960 μF, using a Bio-Rad Gene Pulser, and transfected with 100–250 nM siRNA. For osmolarity experiments, 1 × 10^6^ electroporated cells were seeded in 25 cm^2^ flasks, or 2 × 10^4^ cells seeded on glass cover slips.

#### 4.2.4. RNA Extraction, cDNA Synthesis and Reverse Transcription PCR (RT-PCR)

Isolation of total RNA, synthesis of cDNA and PCR reactions were performed as previously described [20,53]. PCR reactions were performed using specific primers and cycling protocols (Table 2). Gel documentation and densitometry analysis were performed using Image Lab Software version 5.2 (Bio-Rad Laboratories), with correction for loading by the housekeeping gene glyceraldehyde-3-phosphate dehydrogenase (*Gapdh*).

#### 4.2.5. Quantitative PCR (qPCR)

RNA extraction and cDNA synthesis were performed as described for the RT-PCR protocol. Primers were designed using PrimerBLAST software (NCBI) and/or taken from the literature. The primers were obtained from Eurofins Genomics (Table 3) and tested for primer efficiency using serially diluted cDNA (see Table 3). Quantitative PCR (qPCR) was performed essentially as described [20,54] in a StepOnePlus Real-Time PCR System (Applied Biosystems), using KAPA SYBR FAST qPCR Master Mix Universal and High ROX Reference Dye (Roche). The cycling conditions were activation at 95 °C for 5 min, followed by 40 cycles (42 cycles for *Aqp2*) of 95 °C for 3 s, and of 60 °C (62 °C for *Aqp2*) for 30 s, with melt curve analysis to check amplification specificity. Gene expression levels were calculated according to the 2^−ΔCq^ method relative to the sample with the highest expression (minimum Cq) [55]. The data obtained were normalized to the expression of two stable reference genes: *Gapdh* and β-actin (*Actb*).

#### 4.2.6. Measurements of Transepithelial Electrical Resistance (TEER) of Cell Monolayers

Cells (2 × 10^4^) were plated in transwell filters for 24-well plates with 0.4-μm pore size and 0.33-cm^2^ surface area (cat. # 3470, Costar Transwell-Clear, Corning, Wiesbaden, Germany). After two days, transepithelial electrical resistance (TEER) was measured daily, using an epithelial volt-ohm meter EVOM with an STX2 electrode (World Precision Instruments, Friedberg, Germany), at room temperature. Reported resistance readings were corrected for background by subtracting the resistance of empty cell-free inserts in culture medium of 140.3 ± 4.0 Ω (means ± SEM; *n* = 17). Treatments were started 14–15 days after seeding, when TEER reached a stable value of 284.3 ± 15.9 Ω × cm2 (means ± SEM; *n* = 17). LPS and/or ddAVP were then added to both the apical (upper chamber, 300 μL medium) and basolateral (lower chamber, 1000 μL medium) compartments and incubated for 18 h at 37 °C, prior to measurements of *Lcn2* expression and secretion.

#### 4.2.7. Determination of LCN2 Expression and Secretion

LCN2 secretion by mCCD(cl.1) cells was detected by immunoblotting. Media from the apical and basolateral compartments of transwell filters were concentrated at room temperature to a final volume of 100–150 μL, using Vivaspin 500 centrifugal concentrators (10 kDa molecular weight cut-off; cat. # VS0102, Sartorius), corrected for volume differences and used for immunoblotting. To determine cellular LCN2 expression, cells grown on filters were lysed in 30 μL/well of lysis buffer (25 mM Tris pH 7.4, 2 mM Na_3_VO_4_, 10 mM NaF, 10 mM Na_4_P_2_O_7_, 1 mM ethylenediaminetetraacetic acid (EDTA), 1 mM ethylene glycol-bis(β-aminoethyl ether)-N,N,N′,N′-tetraacetic acid (EGTA), 1% Nonidet P-40) containing protease inhibitors. For *Lcn2* expression by RT-PCR, cells grown on transwell filters were washed once with phosphate buffered saline (PBS), and collected by scraping in 100 μL PBS, using a rubber policeman.

#### 4.2.8. Isolation of Plasma Membrane Enriched Microsomes

An amount of 2 × 10^6^ mCCD(cl.1) cells were seeded into 175 cm^2^ culture flasks and grown for 24 h in standard culture medium before osmotic challenge for 72 h. Plasma membranes were obtained by differential ultracentrifugation at 4 °C. Cells were homogenized by nitrogen pressure cavitation in a Parr Instruments 45 mL cell disruption vessel (Moline, IL, USA) for 2 min at 1000 p.s.i. To remove unbroken cells, nuclei, large debris and mitochondria, homogenate was centrifuged at 8000× *g* for 20 min, and the resulting supernatant was centrifuged for 45 min at 35,000× *g,* to yield a microsomal fraction in the pellet, before being resuspended and supplemented with protease inhibitors. Plasma membrane purity was verified by immunoblotting and showed about a 10-fold enrichment of Na^+^,K^+^-ATPase (Figure 1C).

#### 4.2.9. Isolation of Nuclei

An amount of 4 × 10^5^ cells were plated into 75 cm^2^ culture flasks and grown for 48 h, to reach a confluency of ~80%, before medium was replaced by 300–500 mosmol/L media and incubated for an additional 24 h. Nuclei and post-nuclear supernatant were separated as reported elsewhere [56]. All steps were performed at 4 °C. Cells were swelled in hypotonic buffer A (10 mM HEPES-KOH, pH 7.9, 1.5 mM MgCl_2_, 10 mM KCl, 0.5 mM dithiothreitol (DTT), 0.05% Nonidet P-40) containing protease inhibitors, Tenbroeck homogenized, and nuclei were pelleted (220× *g*, 5 min). Nuclei were washed with buffer A + 0.3% Nonidet P-40, to remove cytoplasmic contaminants, and pelleted (600× *g*, 5 min). Nuclei were resuspended in 0.25 M sucrose/10 mM MgCl_2_, layered over 0.35 M sucrose/0.5 mM MgCl_2_ and centrifuged at 1430× *g* for 5 min. Purified nuclei were stored in buffer B (20 mM HEPES-KOH, pH 7.9, 25% glycerol, 420 mM NaCl, 1.5 mM MgCl_2_, 0.2 mM EDTA, 0.5 mM DTT, 0.05% Nonidet P-40), containing protease inhibitors. Samples were sonicated using a Branson Digital Sonifier (Danbury, CT, USA) (3 × 5 s at 20% output) prior to protein determination by the Bradford method [57], using bovine serum albumin (BSA) as a standard.

#### 4.2.10. Immunoblotting

Sodium dodecyl sulfate polyacrylamide gel electrophoresis (SDS-PAGE) and immunoblotting were essentially performed according to standard procedures of wet transfer, as described earlier [14], with the exception of LCN2 immunoblots, which were electroblotted by rapid semi-dry transfer (Bio-Rad Laboratories Trans-Blot Turbo). Cells were washed in PBS, scraped and homogenized by sonication in an isosmotic sucrose buffer. Protein concentration was determined by the Bradford method [57]. Equal amounts of protein (5–30 μg) were subjected to SDS-PAGE and immunoblotted. Primary antibodies and their dilutions are listed in Table 1. Horseradish peroxidase-conjugated secondary antibodies were purchased from Santa Cruz Biotechnology, Inc. or GE Healthcare (Heidelberg, Germany or Schwerte, Germany) and used at 1:2500–1:30,000 dilutions. Densitometry analysis was performed using ImageJ software (https://imagej.net/) [58].

#### 4.2.11. Immunofluorescence Staining and Microscopy

mCCD(cl.1) cells (0.5–2.0 × 10^4^ cells / well) were plated on glass coverslips and cultured for 24–72 h, to reach a confluence of 40%–50% prior to treatments.

For surface staining (at 4 °C), cells were blocked with 1% BSA-PBS for 1 h, followed by incubation with a rabbit polyclonal IgG antibody against the N-terminus of rodent anti-SLC22A17 (1:100 dilution in 1% BSA-PBS = 10 μg/mL) for 2 h. Cells were then incubated with secondary Alexa Fluor 488-conjugated goat anti-rabbit IgG (1:600; cat. # A-11008, Thermo Fisher) for 1 h, and fixed with 4% paraformaldehyde (PFA) in PBS for 30 min at room temperature. Nuclei were counterstained with 0.8 μg/mL Hoechst 33342 for 5 min. Coverslips were embedded with DAKO fluorescent mounting medium, and images were acquired as described elsewhere [59].

For total cellular SLC22A17 staining at room temperature (LPS experiments), cells were fixed with 4% PFA in PBS for 30 min, permeabilized with 1% SDS in PBS for 15 min, and blocked with 1%–2% BSA-PBS for 30–40 min. N-terminal rodent anti-SLC22A17 (1:100–1000 dilution in 1% BSA-PBS) or LCN2 polyclonal rabbit IgG (Abcam Ab63929: 1:250–1000 dilution in 2% BSA-PBS) were incubated for 2 h. Subsequent steps were identical to surface staining procedures. For quantitative determination of Alexa Fluor 488-conjugated SLC22A17 or LCN2 in mCCD(cl.1) cells, 200 to 500 cells, in six to twelve microscopic fields, were analyzed for each experiment with the MetaMorph software (ICIT, Frankfurt am Main, Germany) [59].

#### 4.2.12. Statistics

Unless otherwise indicated, the experiments were repeated at least three times with independent cultures. Means ± SEM are shown. Statistical analysis, using unpaired Student’s *t*-test, was carried out with GraphPad Prism v. 5.01 (GraphPad Software Inc., San Diego, CA, USA). For more than two groups, one-way ANOVA with Tukey or Dunnett’s *post-hoc* test, assuming equality of variance, was applied. Results with *p* < 0.05 were considered statistically significant. The dose–response curve of the effect of LPS on LCN2 secretion was fitted with the Sigma Plot 12.5 spreadsheet program, assuming a sigmoidal function and using the Hill equation. Significance levels were labelled in the Figures as follows: * = *p* < 0.05; ** = *p* < 0.01; *** = *p* < 0.001.

## 5. Conclusions

In summary (see Figure 7), the present study unveils a signaling mechanism in the cultured CCD cell line mCCD(cl.1) involved in the regulation of the expression of the LCN2 receptor SLC22A17, by hyperosmolarity via NFAT5, and by AVP via CREB, which parallels *Aqp2* regulation, suggesting SLC22A17 plays a role in adaptation to osmotic stress. In contrast, the ligand of SLC22A17, LCN2 is downregulated by hyperosmolarity and AVP in a NFAT5-dependent, CREB-independent manner. Rather, PKA-dependent posttranslational control of LCN2 occurs via ill-defined mechanisms. We speculate that LCN2 downregulation may prevent the increased proliferation and permanent damage of osmotically stressed cells.

## Figures and Tables

**Figure 1 ijms-20-05398-f001:**
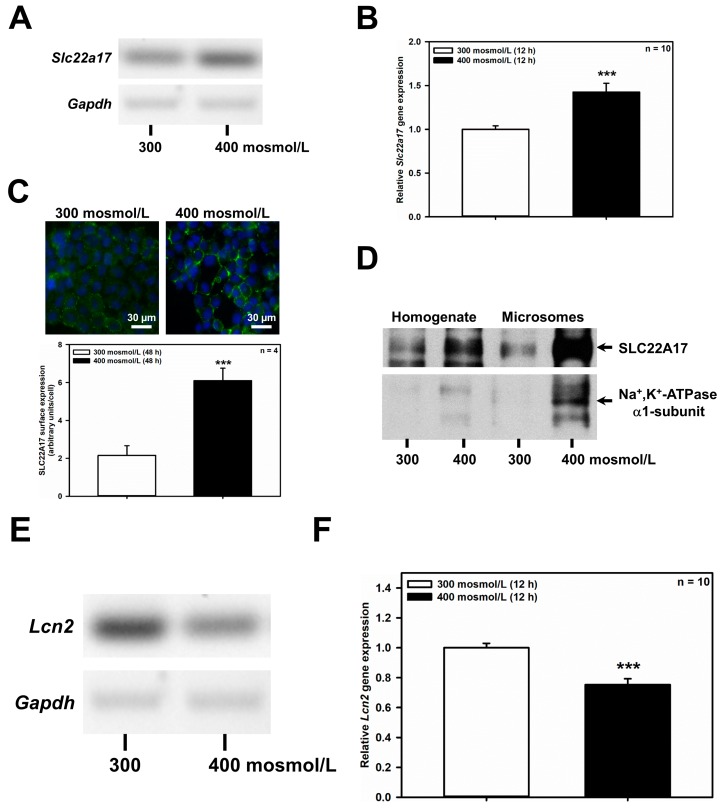
Hyperosmolarity increases *Slc22a17*/SLC22A17 expression and decreases *Lcn2* expression in mCCD(cl.1) cells. (**A**) RT-PCR analysis of *Slc22a17* and *Gapdh* mRNA in mCCD(cl.1) cells exposed to 300 mosmol/L (normosmolarity) or 400 mosmol/L (hyperosmolarity) for 12 h. The experiment is similar to three others. (**B**) Expression levels of *Slc22a17* mRNA by qPCR in mCCD(cl.1) cells exposed to norm- or hyperosmotic media for 12 h. Means ± SEM of 10 experiments are shown. Data normalized to the expression of *Gapdh* and *Actb* show relative expression levels of *Slc22a17* under hyperosmotic conditions, where expression at 300 mosmol/L is set to 1.0. Statistics compare hyper- to normosmolarity by unpaired *t*-test. (**C**) Surface expression of SLC22A17 in mCCD(cl.1) cells exposed to norm- or hyperosmotic media for 48 h. SLC22A17 is detected by live immunofluorescence microscopy of non-fixed and non-permeabilized cells with a SLC22A17 antibody directed against the extracellular N-terminus. Hoechst 33342 counterstains nuclei. Means ± SEM of four experiments and comparison of the two osmotic conditions by unpaired *t*-test are shown. a.u. = arbitrary units. (**D**) Immunoblotting of homogenate and microsomes enriched in plasma membranes from mCCD(cl.1) cells grown for 72 h in norm- or hyperosmotic media. SLC22A17 is at the expected molecular mass of ~62 kDa. The α1-subunit of Na^+^, K^+^-ATPase, a plasma membrane marker, is enriched in microsomes and upregulated in cells exposed to hyperosmolarity. The experiment is representative of three similar ones. (**E**) RT-PCR analysis of *Lcn2* and *Gapdh* mRNA in mCCD(cl.1) cells exposed to 300–400 mosmol/L media for 12 h. The experiment is typical of three similar ones. (**F**) Expression levels of *Lcn2* mRNA by qPCR in mCCD(cl.1) cells exposed to 300–400 mosmol/L media for 12 h. Means ± SEM of 10 experiments are shown. Data normalized to the expression of *Gapdh* and *Actb* show relative expression levels of *Lcn2* under hyperosmotic conditions, where expression at 300 mosmol/L is set to 1.0. Statistics compare the two osmotic conditions by unpaired *t*-test. For a definition of asterisks, see “statistics” in the Methods.

**Figure 2 ijms-20-05398-f002:**
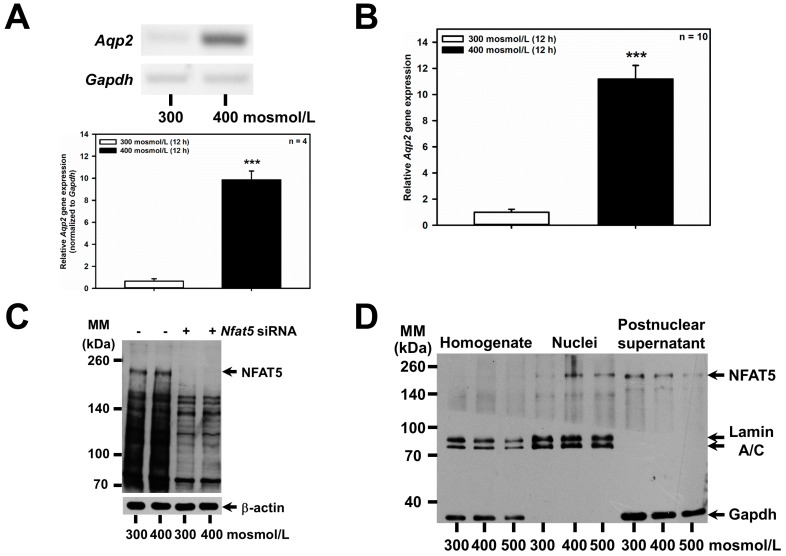
*Aqp2* upregulation induced by hyperosmolarity is mediated by NFAT5 in mCCD(cl.1) cells. (**A**) Expression levels of *Aqp2* mRNA by RT-PCR in mCCD(cl.1) cells exposed to 300–400 mosmol/L for 12 h. *Aqp2* mRNA expression was normalized to *Gapdh*, and expression levels at 300 mosmol/L are set to 1.0. Data show means ± SEM of four experiments. Statistical analysis compares hyper- to normosmolarity by unpaired *t*-test. (**B**) Expression levels of *Aqp2* mRNA by qPCR in mCCD(cl.1) cells exposed to 300–400 mosmol/L. Means ± SEM of 10 experiments are shown. Data normalized to the expression of *Gapdh* and *Actb* show relative expression levels of *Aqp2* under hyperosmotic conditions, where expression at 300 mosmol/L is set to 1.0. Statistics compare hyper- to normosmolarity by unpaired *t*-test. (**C**) mCCD(cl.1) cells were transfected with control or *Nfat5* siRNA by electroporation and cultured, as described above, prior to a medium change to 300 and 400 mosmol/L for an additional 24 h, and immunoblotting. β-actin serves as a loading control. The blot is typical of three similar ones. MM = molecular mass. (**D**) Distribution of NFAT5 in homogenate, nuclei and postnuclear supernatant of mCCD(cl.1) cells exposed to 300–500 mosmol/L for 24 h prior to homogenization, subcellular fractionation and immunoblotting of NFAT5. Lamin A/C and GAPDH are markers of nuclei and cytosol, respectively. The immunoblot is typical of three different blots. MM = molecular mass. (**E**,**F**) mCCD(cl.1) cells are transfected with siRNA against *Nfat5* or control siRNA by electroporation. Then cells are seeded in 25 cm^2^ flasks and grown for 24 h, prior to a medium change to 300 or 400 mosmol/L and incubation for an additional 6–12 h. Expression levels of *Nfat5* (**E**) and *Aqp2* mRNA (**F**) are determined by qPCR, as described above. Statistical analysis shows means ± SEM of 10–13 experiments and compares experimental conditions by one-way ANOVA. n.s. = not significant. For a definition of asterisks, see “statistics” in the Methods.

**Figure 3 ijms-20-05398-f003:**
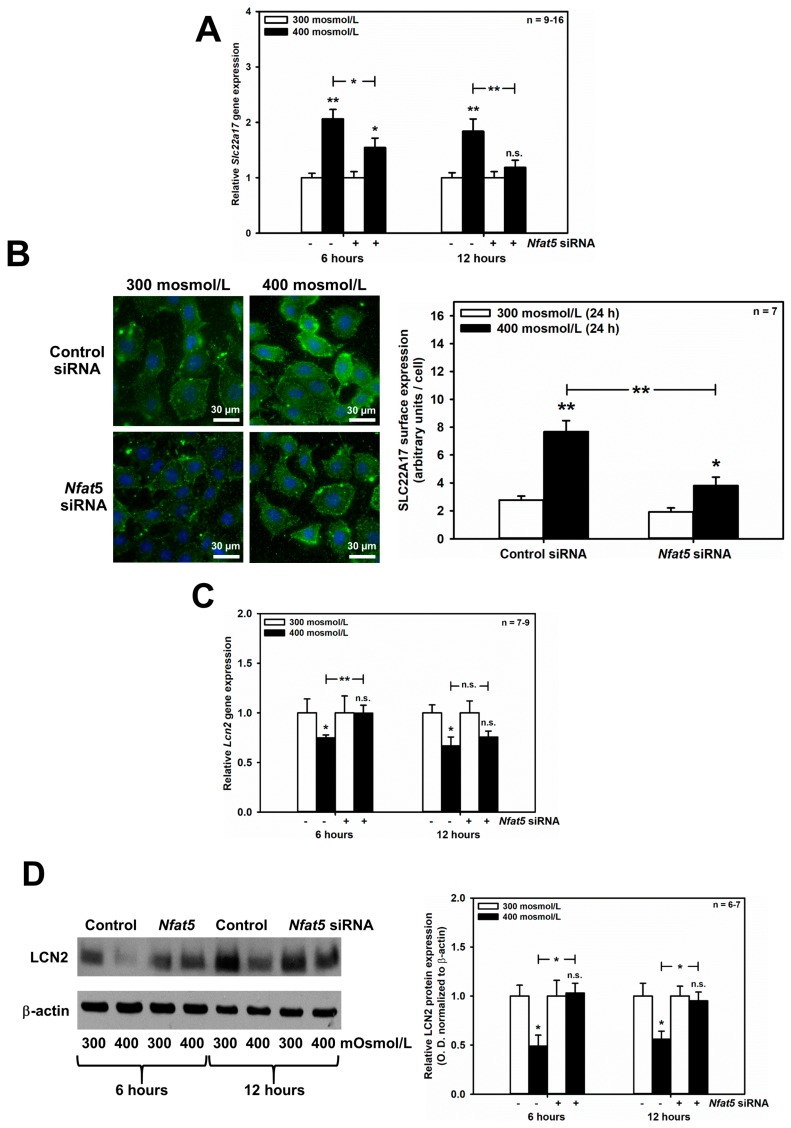
NFAT5 mediates the effects of hyperosmolarity on *Slc22a17*/SLC22A17 and *Lcn2*/LCN2 expression in mCCD(cl.1) cells. (**A**,**C**) mCCD(cl.1) cells were transfected with control or *Nfat5* siRNA by electroporation, and cultured as described in the Methods, prior to a medium change to 300 or 400 mosmol/L and incubation for additional 6–12 h. Expression levels of *Slc22a17* (**A**) and *Lcn2* (**C**) mRNA were detected by qPCR, as described in Figure 1. Statistical analysis shows means ± SEM of 7–16 experiments and compares experimental conditions by one-way ANOVA. n.s. = not significant. (**B**) Surface expression of SLC22A17 was determined in mCCD(cl.1) cells, transfected with siRNA against *Nfat5*, or control siRNA, by electroporation. Cells are exposed to 300 or 400 mosmol/L for an additional 24 h. Plasma membrane SLC22A17 is detected, as described in Figure 1. Hoechst 33342 counterstains nuclei. Statistical analysis shows means ± SEM of seven experiments and compares the experimental conditions by one-way ANOVA. n.s. = not significant. (**D**) Immunoblotting of cellular LCN2 protein in mCCD(cl.1) cells, transfected with siRNA against *Nfat5*, or control siRNA, by electroporation. Incubation of cells occurs in 300 or 400 mosmol/L media for an additional 6–12 h, prior to immunoblotting. Data show cellular LCN2 protein expression normalized to β-actin as means ± SEM of 6–7 experiments, where expression at 300 mosmol/L is set to 1.0. Statistical analysis compares the experimental conditions by one-way ANOVA. n.s. = not significant. For a definition of asterisks, see “statistics” in the Methods.

**Figure 4 ijms-20-05398-f004:**
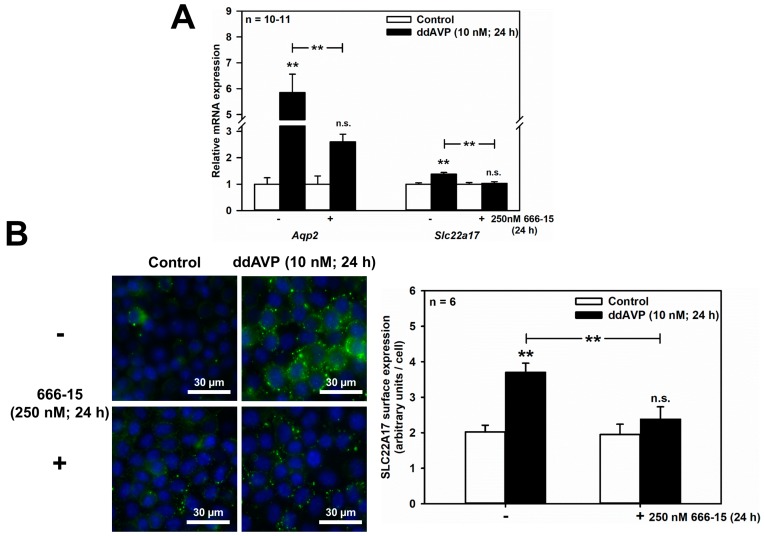
ddAVP increases *Slc22a17*/SLC22A17 expression through CREB activation in mCCD(cl.1) cells. (**A**) Expression levels of *Aqp2* and *Slc22a17* mRNA by qPCR, in cells treated with ddAVP ± the CREB inhibitor 666-15, and analyzed and calculated as described in Figure 1. Data are means ± SEM of 10–11 experiments. Statistical analysis compares relative expression levels of *Aqp2* or *Slc22a17* in controls and ddAVP-treated cells ± 666-15 by one-way ANOVA. n.s. = not significant. (**B**) Surface expression of SLC22A17 in cells treated with ddAVP ± 666-15. Staining of non-fixed and non-permeabilized cells grown on glass coverslips was performed using a SLC22A17 antibody, directed against the extracellular N-terminus. Hoechst 33342 counterstains nuclei. Statistical analysis shows the means ± SEM of six experiments, and compares the four experimental conditions by one-way ANOVA. n.s. = not significant. For a definition of asterisks, see “statistics” in the Methods.

**Figure 5 ijms-20-05398-f005:**
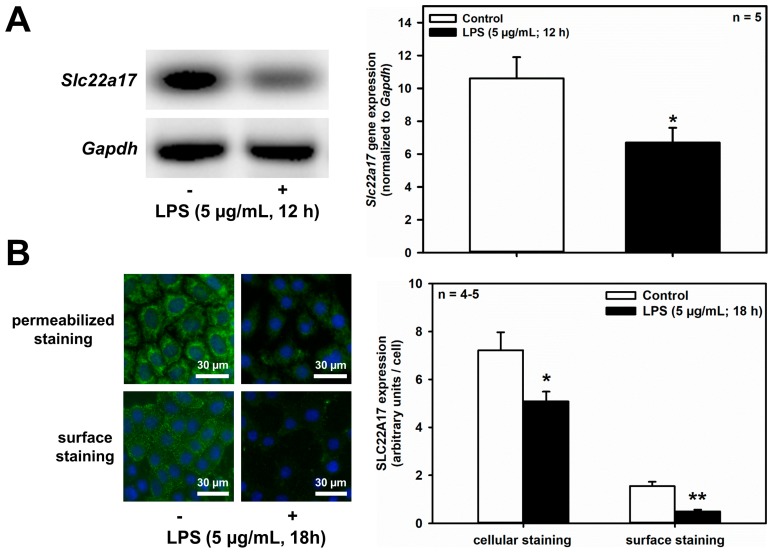
LPS decreases *Slc22a17*/SLC22A17 expression and increases *Lcn2*/LCN2 expression in mCCD(cl.1) cells. (**A**) Expression levels of *Slc22a17* mRNA by RT-PCR, in mCCD(cl.1) cells treated with 5 μg/mL lipopolysaccharides (LPS) for 12 h. *Slc22a17* mRNA expression was normalized to *Gapdh*. Data show means ± SEM of five experiments. Statistical analysis compares the effects of control versus LPS by unpaired *t*-test. (**B**) Expression of SLC22A17 in mCCD(cl.1) cells treated with 5 μg/mL LPS for 18 h was detected by immunofluorescence microscopy of permeabilized and non-permeabilized (“surface staining”) cells, using a SLC22A17 antibody directed against the extracellular N-terminus. Hoechst 33342 counterstains nuclei. Statistical analysis shows means ± SEM of 4–5 experiments and comparison of the two conditions by unpaired *t*-test. (**C**) Expression levels of *Lcn2* mRNA by RT-PCR in mCCD(cl.1) cells treated with LPS. *Lcn2* mRNA expression was normalized to *Gapdh*. Data show means ± SEM of six experiments. Statistical analysis compares the effects of control versus LPS by unpaired *t*-test. (**D**) Effect of LPS on expression of LCN2 protein in mCCD(cl.1) cells. Cellular LCN2 protein expression was normalized to β-actin. Data show means ± SEM of four experiments. Statistical analysis determines the effect of LPS on cellular LCN2 protein using unpaired *t*-test. (**E**) Effect of LPS on cellular expression and secretion of LCN2 protein in mCCD(cl.1) cell monolayers, grown to confluence on transwell filters. LPS or solvent were applied to both apical and basolateral chambers for 18 h. Cells were lysed, apical and basolateral media were collected and concentrated to the same volume, as described in the Methods. Corresponding corrected volumes of concentrated media were loaded for immunoblotting. Data show cellular LCN2 protein expression normalized to β-actin, as means ± SEM of 9–10 experiments. Statistical analysis determines the effect of LPS on cellular and apically or basolaterally secreted LCN2 protein, using one-way ANOVA. For a definition of asterisks, see “statistics” in the Methods.

**Figure 6 ijms-20-05398-f006:**
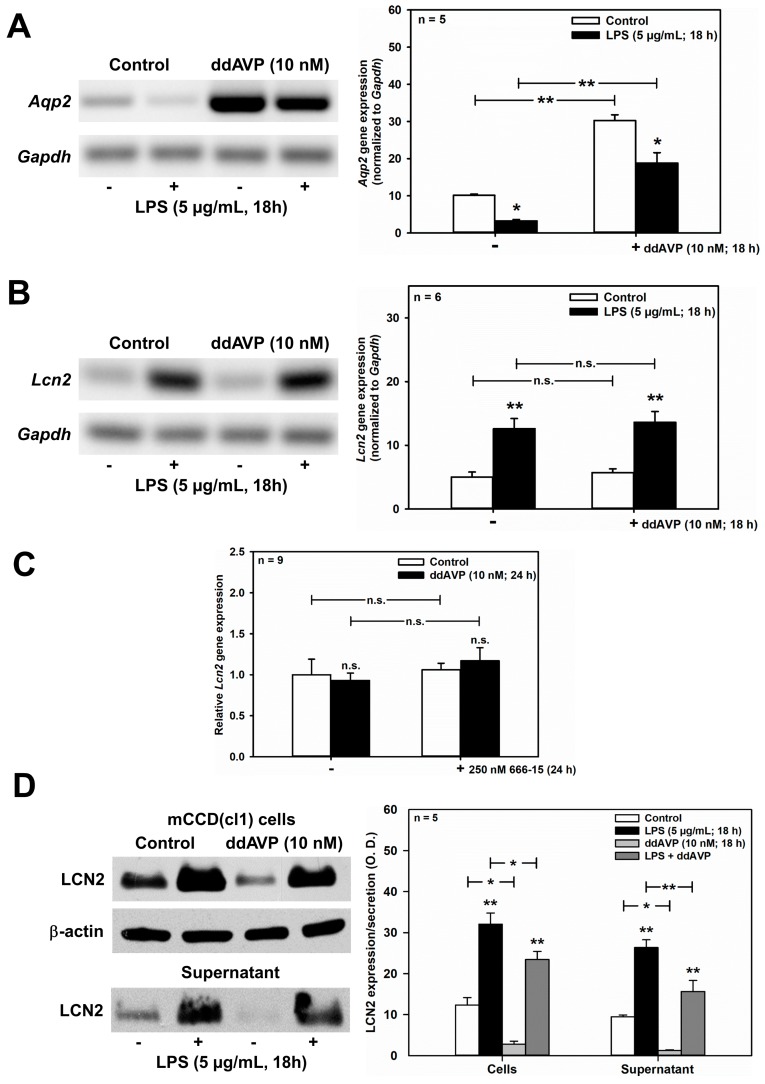
ddAVP decreases *Lcn2*/LCN2 expression through a CREB-independent, posttranslational mode of action in mCCD(cl.1) cells. (**A**,**B**) Expression levels of *Aqp2* (**A**) and *Lcn2* (**B**) mRNA by RT-PCR in mCCD(cl.1) cells treated with ddAVP ± LPS. Data are means ± SEM of 5–6 experiments. Statistical analysis compares the expression levels of *Aqp2* or *Lcn2* in controls and ddAVP-treated cells ± LPS, by one-way ANOVA. n.s. = not significant. (**C**) Expression levels of *Lcn2* mRNA by qPCR in mCCD(cl.1) cells treated with ddAVP ± 666-15. Data are means ± SEM of nine experiments. Statistical analysis compares relative expression levels of *Lcn2* in controls and ddAVP-treated cells ± 666-15, by one-way ANOVA. n.s. = not significant. (**D**) Expression and secretion of LCN2 in mCCD(cl.1) cells treated with LPS, in the absence or presence of ddAVP. Media (supernatant), treated as described in Figure 5E, were immunoblotted. Cellular LCN2 protein expression was normalized to β-actin. Data are means ± SEM of five experiments. Statistical analysis compares expression levels of cellular and secreted LCN2 in controls and ddAVP-treated cells ± LPS, by one-way ANOVA. (**E**) Expression of LCN2 in mCCD(cl.1) cells treated with ddAVP for 6 h ± cycloheximide, a blocker of translational elongation, was detected by immunofluorescence microscopy of permeabilized cells. Hoechst 33342 counterstains nuclei. Statistical analysis shows means ± SEM of four experiments and comparison of the conditions ± ddAVP by unpaired *t*-test. n.s. = not significant. For a definition of asterisks, see “statistics” in the Methods.

**Figure 7 ijms-20-05398-f007:**
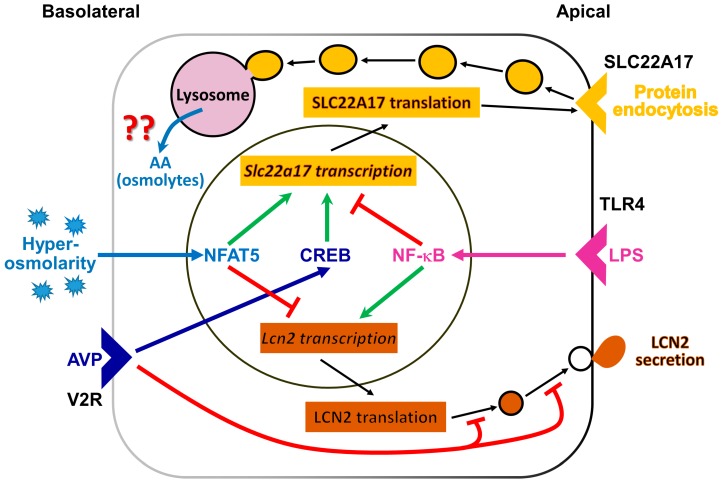
Regulation of lipocalin-2 receptor (SLC22A17) and lipocalin 2 (LCN2) abundance in the renal cortical collecting duct cell line mCCD(cl.1). Several extracellular stimuli control SLC22A17and LCN2 abundance, by acting on *Slc22a17* and *Lcn2* gene transcription or posttranslational *Lcn2* expression. Shown are pathways mediated by arginine vasopressin (AVP), binding to the arginine vasopressin receptor-2 (V2R) (dark blue), hyperosmolarity/-tonicity (light blue) and/or inflammation (magenta). AVP stimulation (dark blue arrow) results from adenylyl cyclase activation, increased intracellular cAMP concentration, and protein kinase A (PKA) activation. PKA, in turn, increases cAMP-responsive element binding protein (CREB) activity. NF-κB activation (magenta arrow) occurs by inflammatory stimuli, such as lipopolysaccharide (LPS) via toll-like receptor 4 (TLR4). After longer periods of hypertonic challenge, the nuclear factor of activated T-cells 5 (NFAT5; also known as TonEBP or OREBP) is activated. Apically expressed SLC22A17 promotes protein endocytosis, possibly to provide amino acids (AA) and/or osmolytes for osmotolerance, but this remains unproven. Cells secrete LCN2 to combat bacterial infections and/or promote proliferation after epithelial damage. ↑ (green) = stimulation; 
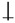
 (red) = inhibition. For further details, see text.

**Table 1 ijms-20-05398-t001:** Primary antibodies.

Immunogen	Host Species	Manufacturer	Catalog #	Application	Dilution
Rat LCN2	goat	RD Biosystems	AF3508	IB *	1:200–1:300
Mouse LCN2	rabbit	Abcam	Ab63929	IF *	1:250–1:1000
Rat SLC22A17 (C-terminus)	rabbit	[14]	Ig-1086	IB	1:100–1:2000
Rat SLC22A17 (N-terminus)	rabbit	[14]	Ig-1095	IF	1:100–1:1000
Human b-actin	mouse	[14]	A5316	IB	1:20,000
Human GAPDH (14C10)	mouse	Sigma-Aldrich	2118	IB	1:20,000
Human Na^+^,K^+^- ATPase a1-subunit	rabbit	Cell Signaling	3010S	IB	1:250–1:500
Human Lamin A/C (4C11)	mouse	Cell Signaling	4777	IB	1:20,000
Human NFAT5	rabbit	[24]	N/A	IB	1:6000
Human IkB-a	rabbit	Santa Cruz	Sc-371	IB	1:1000

* IB = immunoblotting; * IF = immunofluorescence.

**Table 2 ijms-20-05398-t002:** Protocols for reverse transcription PCR.

	*Gapdh*	*Aqp2*	*Lcn2*	*Slc22a17*
**Accession number**	NM_001289726.1	NM_009699.3	NM_008491.1	NM_021551.4
**Forward primer (5′-3′)**	AGGGCTCATGACCACAGT	TGGCTGTCAATGCTCTCCAC	CCACCACGGACTACAACCAG	CAGCCACCTCCTAACCGCTGTG
**Reverse primer (5′-3′)**	TGCAGGGATGATGTTCTG	GGAGCAGCCGGTGAAATAGA	AGCTCCTTGGTTCTTCCATACA	CTCCCACTAGGCTCAAAGGCTGCT
**Reference**	NCBI Primer-BLAST	[53]	[39]	[39]
**Activation**	5 min 95 °C	5 min 95 °C	5 min 95 °C	5 min 95 °C
**Cycle number**	18–22	31–33	20–25	27–29
**Denaturation**	30 s 94 °C	30 s 94°C	30 s 94 °C	30 s 94 °C
**Annealing**	30 s 60 °C	30 s 62 °C	30 s 60 °C	30 s 60 °C
**Extension**	30 s 72 °C	30 s 72 °C	30 s 72 °C	30 s 72 °C
**Final Extension**	7 min 72 °C	7 min 72 °C	7 min 72 °C	7 min 72 °C
**PCR product (bp)**	112	200	100	86

**Table 3 ijms-20-05398-t003:** qPCR primers.

Gene-Name (Accession Number)	Forward (5′-3′)	Reverse (5′-3′)	Reference	Amplicon Size (bp)	*Efficiency(%)*
***Actb*** ***(NM_007393.5)***	CGTGCGTGACATCAAAGAGAA	GGCCATCTC CTGCTCGAA	[20]	102	102
***Gapdh*** ***(NM_001289726.1)***	CGGCCGCATCTTCTTGTG	CCGACCTTCACCATTTTGTCTAC	[20]	100	100
***Lcn2*** ***(NM_008491.1)***	CCACCACGGACTACAACCAG	AGCTCCTTGGTTCTTCCATACA	[20]	98	98
***Slc22a17*** ***(NM_021551.4)***	CAGCCACCTCCTAACCGCTGTG	CTCCCACTAGGCTCAAAGGCTGCT	[20]	110	110
***Aqp2*** ***(NM_009699.3)***	TGGCTGTCAATGCTCTCCAC	GGAGCAGCCGGTGAAATAGA	[53]	92	92

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
