# Peer review of "Inverse Regulation of Lipocalin-2/24p3 Receptor/SLC22A17 and Lipocalin-2 Expression by Tonicity, NFAT5/TonEBP and Arginine Vasopressin in Mouse Cortical Collecting Duct Cells mCCD(cl.1): Implications for Osmotolerance"

_ijms, 2019, doi:10.3390/ijms20215398_

Round 1

Reviewer 1 Report

The manucript by Probst et all addresses regulation of expression of lipocalin-2/24p3 2 receptor/Slc22a17 and lipocalin-2 in mouse cortical collecting duct cells. This is continuation  of their previous study that demonstrated similar mechanisms in inner medula collecting duct cell line. The results are presented clearly and the paper is well written. However, novelty of the paper is not high enough.

The major limitation of the paper is that the authors performed all experiments using a cell line only. No relevance to in vivo situation is provided. The paper addresses only the regulation of ipocalin-2/24p3 2 receptor/Slc22a17 expression without giving any insight into the function of the receptor. Authors should  improve the novelty by studying the role of the protein, at least using in vitro model.  Minor points:

1. RT-PCR data should be quantified (Fig. 1A and so on). 2. It should be explained, why was 400 mosmol/l selected. 3. Immunostaining images are of very poor quality and must be improved. 4. Fig.2. In the text it is stated 500mosmol/l (page 5, line 149), in the Figure legend - 400. Please, correct. 5. In the Figure legend 2C it is stated that this is qPCR data, whereas western blotting image is show in 2C. 6. Figure 2D. How can the suthoirs explain that no Nfat5 is detected in homogenates? 7. Figure 2E. Why there is no increase of Nfat5 mRNA is observed in siRNA-treated cells? 8. 5µg/ml LPS concentration used in the study is too high and not physiological. LPS effects should be studied at lower concentrations, otherwise they probably have no physiological relevance. In addition, at high concentration LPS can induce non-canonical TLR4-independent inflammasome pathway. 9. It is not stated, how fluorescent images were quantified. The data should be confirmed by other methods (western, FACS).

Author Response

Point-by-point answer to the Reviewers

Reviewer 1

We thank the Reviewer for her/his detailed and accurate comments that have helped to improve the manuscript.

The manucript by Probst et all addresses regulation of expression of lipocalin-2/24p3 2 receptor/Slc22a17 and lipocalin-2 in mouse cortical collecting duct cells. This is continuation  of their previous study that demonstrated similar mechanisms in inner medula collecting duct cell line. The results are presented clearly and the paper is well written. However, novelty of the paper is not high enough.

We think we have not made clear enough in the introduction and the discussion how important and novel the data are. First of all, the lipocalin-2 receptor is expressed in the kidney medulla where high osmolarity prevails. Without an adaptive response, cells would normally not survive in this harsh environment thus the molecular mechanisms underlying osmotolerance is vital to our understanding of kidney function. Loss of these cells when osmotolerance is perturbed would lead to disruption of urine concentration, major loss of fluid from the body resulting in death. Cell survival occurs via osmotic adaptation that is mediated by Nfat5. The fact that both Nfat5 and the urine concentrating hormone arginine vasopressin (AVP) regulates the lipocalin-2 receptor is per se novel and significantly advances the field of osmotolerance and renal medullary cell survival as well as further elucidating physiological roles of the receptor.

When the lipocalin-2 receptor was cloned in 2005 (published in Cell), its role as a receptor involved in cancer was assumed (ref. 13). Moreover, novel roles in cancer proliferation and metastasis via iron transport are currently discussed (Jung M. et al. Int J Mol Sci. 2019 Jan; 20(2): 273). We were the first to identify the lipocalin-2 receptor in the kidney (ref. 14), where it plays a role in inflammation (ref. 15). We have proven that the receptor binds lipocalin-2 (ref. 21) and recently provided evidence for its survival role during ascending urinary bacterial infection in renal collecting ducts in a cell culture model (ref. 42). However, understanding of the physiological function of the receptor in the renal medulla, where it is highly expressed, is scant.

Functional aspects in this context are of course important, and we have worked on that for years. For instance, in 2015 we have generated a renal (whole nephron- as well as collecting duct-specific) lipocalin-2 receptor ko mouse. Yet the mouse, though viable, has difficulties to reproduce, and so far, we have been unable to generate a colony. Other functional approaches are elaborated below.

The major limitation of the paper is that the authors performed all experiments using a cell line only. No relevance to in vivo situation is provided.

As mentioned above, the only clear-cut way for in vivo testing is the nephron-specific ko mouse, which is currently not feasible. Speculations on the role of the in vivo role of the receptor receptor have already been provided in the last paragraph of the discussion. Moreover, the links to osmolarity and AVP that are physiological regulators of kidney function are obvious.

The paper addresses only the regulation of ipocalin-2/24p3 2 receptor/Slc22a17 expression without giving any insight into the function of the receptor.

The function of the receptor has been described in previous publications (Refs. 14, 15, 21, 42). A function in the context of Nfat5 and AVP regulation is not feasible for a revision and is a project of its own. The key issue is to look for a specific end-point. For instance, if you knockout Nfat5 and look for the end-point cell viability, you will definitely see decreased survival. But Nfat5 activates a plethora of adaptive processes (see ref. 9). These processes may compensate for the loss of lipocalin-2 receptor-mediated osmotolerance, if you knockout the receptor. What we are currently investigating are end-points of intracellular osmolarity and molecular crowding using specific fluorescent tools for those end-points. We aim to overexpress the receptor in a heterologous system. The complexity of the issue is further enhanced by the fact that the receptor has several splice variants, including the A- and B-form, that show different functions regarding osmolarity and cell fate (unpublished results). We then then need to identify among the various physiological protein ligands of the receptor the ligand that will generate most osmolytes and then test osmolarity, molecular crowding (and perhaps cell fate). By using this approach we may be able to bypass Nfat5 to specifically focus on the role of the lipocalin-2 receptor in osmotolerance. Needless to say, we are still establishing and optimizing the methodology, and we don’t think that that work belongs to the revision of this manuscript.

Authors should  improve the novelty by studying the role of the protein, at least using in vitro model.

See above.

Minor points:

RT-PCR data should be quantified (Fig. 1A and so on).

Quantification has been performed (Figures 2A, 5A, 5C, 6A, 6B). However, since all PCR data were complemented (and confirmed) by qPCR, the PCR data without quantification (Figures 1A, 1E) only exemplify the observed pattern.

It should be explained, why was 400 mosmol/l selected.

The reason has been explained (Introduction: lines 59-62). Yet for better clarification and for readers who are not renal physiologists, a sentence was added in the Methods (lines 434-436 of revised manuscript).

Immunostaining images are of very poor quality and must be improved.

PDF images have been improved and replaced by TIFF/JPG images where qualitative rather than quantitative aspects are required.

Fig.2. In the text it is stated 500mosmol/l (page 5, line 149), in the Figure legend - 400. Please, correct.

This is a misunderstanding. The sentence refers for “up to 500 mosmol/l” to the Figures 2A-2D, not only Figure 2A (Figure 2D also shows 500 mosmol/l). The sentence has been changed (lines 154-156 of revised manuscript).

In the Figure legend 2C it is stated that this is qPCR data, whereas western blotting image is show in 2C.

Thank you. The obvious mistake has been corrected.

Figure 2D. How can the suthoirs explain that no Nfat5 is detected in homogenates?

Nfat5 is detected in homogenates, although weakly (Nfat5 is enriched in respective fractions)! A short-term exposure was selected to show the clear shifts/translocation in nuclei (increase) and post-nuclear supernatants (decrease) at high osmolarity. A long-term exposure (attached) shows the Nfat5/TonEBP band in homogenate.

Figure 2E. Why there is no increase of Nfat5 mRNA is observed in siRNA-treated cells?

If Nfat5 mRNA is silenced with Nfat5 siRNA, the increase of Nfat5 induced by hyperosmolarity is abolished.

5µg/ml LPS concentration used in the study is too high and not physiological. LPS effects should be studied at lower concentrations, otherwise they probably have no physiological relevance. In addition, at high concentration LPS can induce non-canonical TLR4-independent inflammasome pathway.

This concentration is saturating with regard to Lcn2 secretion in CD cells (see ref. 42. Suppl. Figure 4B of ref. 42 and suppl. Figure 1A of the revised manuscript). Moreover, this concentration has been used several times in the literature in mouse collecting duct cell lines (Küper C. et al. Am J Physiol Renal Physiol. 2012 Jan 1;302(1):F38-46; Kim D.G. et al. Inflammation. 2016 Feb;39(1):87-95) as well as in primary mouse kidney collecting duct cells to induce Lcn2 expression in a NF-kB-dependent manner (Paragas, N. et al. Nat Med. 2011 Feb; 17(2): 216–222). We have cited these references in the revised manuscript (lines 257-262 references 37-39 of the revised manuscript).

Moreover, we have included data showing that 5 µg/ml LPS activate the canonical TLR4 pathway by rapid IkB-a degradation (suppl. Figure 1B and lines 257-262  in Results), although the non-canonical TLR4 pathway may also be activated as well and acknowledged this fact.

It is not stated, how fluorescent images were quantified. The data should be confirmed by other methods (western, FACS).

The description of quantification has been done (ref. 59). Moreover, we have included a sentence in the Methods to provide more details on quantification (lines 546-548 of the revised manuscript).

All staining data were confirmed by an independent method for expression (mRNA or protein level). Considering that we always have used two different methods to describe our phenomena and that were consistent, we do not see the necessity of using an additional method that does not add further information.

Reviewer 2 Report

The lipocalin-2 receptor, Lcn2-R, (Slc22a17) is expressed in collecting ducts and has been demonstrated to be upregulated ~4-fold by hyperosmolality , 600 mosmol/l, together with ligand (Lcn2) decreased expression/secretion (Betten et al. Cell Communication and Signaling (2018) 16:74) . These previous experiments also demonstrated that hyperosmolarity-induced Wnt/β-catenin signaling. Under physiological conditions and high interstitial osmolarity, Wnt/β-catenin signaling is activated which leads to downregulation of Lcn2 and upregulation of the Lcn2-R. The latter binds proteins in the primary urine with high affinity that are degraded in lysosomes, possibly to generate adaptive osmolytes. A recent study has implicated Lcn2 secretion from collecting duct (CD) α-intercalated cells as a bacteriostatic defense mechanism against pathogenic bacteria in the urine. Secretion was induced by the bacterial wall component lipopolysaccharide (LPS) within the renal collecting duct (CD) via Toll-like cell surface receptor-4 (TLR-4) activation [Paragas N, Kulkarni R, Werth M, Schmidt-Ott KM, Forster C, Deng R, et al. alpha-Intercalated cells defend the urinary system from bacterial infection. J Clin Invest. 2014;124:2963–76.]

. Bacterial infection in an initially hyperosmotic environment induces TLR-4 signaling that counteracts Wnt/β-catenin signaling, leading to disinhibition of Lcn2 expression and secretion and Lcn2 endocytosis leads to death of IMCD cells. (Betten et al. Cell Communication and Signaling (2018) 16:74).

It is proposed, in this new ms, that, similar to Aqp2, hyperosmotic/-tonic media and AVP upregulated Slc22a17 via activation of the transcription factors Nfat5 (TonEBP or OREBP) (Fig 3), and CREB (Fig 4), respectively, and LPS/TLR4 signaling downregulated Slc22a17 ( Fig 5). Conversely, AVP reduced Lcn2 expression and predominantly apical Lcn2 secretion evoked by LPS, but through a posttranslational mode of action that was independent of cAMP signaling (Fig 6).

Comments : in the discussion, the authors are indicating “In support, a putative Nfat5 DNA consensus motif (osmotic response element (ORE) or tonicity-responsive enhancer (TonE)) [44] was identified in the human SLC22A17 promoter sequence ENSR00001455985/14:23820951-23822367 (828AGGAAAATGCCA839), which is compatible with the data obtained in Figures 3A and 3B. It would be worthwhile to test this putative ORE in future experiments by site-directed mutagenesis of the sequence in a Slc22a17luciferase-reporter gene assay.” Same interesting speculation and need to do luciferase-reporter experiments related to “a putative binding site at position -736” and “a putative NFAT5 DNA binding sequence in a region of the human Lcn2 gene (841TGGAAAAAGGCT852), which could explain Lcn2 downregulation by hyperosmolarity” Would you have any experiments here as well as any data on ko or conditional ko experiments related to Lcn2 or Lcn2-R?. A schematic representation of the new findings of the ms similar to Fig 7 of Betten et al 2018 could be helpful.

Author Response

Point-by-point answer to the Reviewers

Reviewer 2

We thank the Reviewer for her/his detailed and kind comments that have helped to improve the manuscript.

The lipocalin-2 receptor, Lcn2-R, (Slc22a17) is expressed in collecting ducts and has been demonstrated to be upregulated ~4-fold by hyperosmolality , 600 mosmol/l, together with ligand (Lcn2) decreased expression/secretion (Betten et al. Cell Communication and Signaling (2018) 16:74) . These previous experiments also demonstrated that hyperosmolarity-induced Wnt/β-catenin signaling. Under physiological conditions and high interstitial osmolarity, Wnt/β-catenin signaling is activated which leads to downregulation of Lcn2 and upregulation of the Lcn2-R. The latter binds proteins in the primary urine with high affinity that are degraded in lysosomes, possibly to generate adaptive osmolytes. A recent study has implicated Lcn2 secretion from collecting duct (CD) α-intercalated cells as a bacteriostatic defense mechanism against pathogenic bacteria in the urine. Secretion was induced by the bacterial wall component lipopolysaccharide (LPS) within the renal collecting duct (CD) via Toll-like cell surface receptor-4 (TLR-4) activation [Paragas N, Kulkarni R, Werth M, Schmidt-Ott KM, Forster C, Deng R, et al. alpha-Intercalated cells defend the urinary system from bacterial infection. J Clin Invest. 2014;124:2963–76.]

. Bacterial infection in an initially hyperosmotic environment induces TLR-4 signaling that counteracts Wnt/β-catenin signaling, leading to disinhibition of Lcn2 expression and secretion and Lcn2 endocytosis leads to death of IMCD cells. (Betten et al. Cell Communication and Signaling (2018) 16:74).

It is proposed, in this new ms, that, similar to Aqp2, hyperosmotic/-tonic media and AVP upregulated Slc22a17 via activation of the transcription factors Nfat5 (TonEBP or OREBP) (Fig 3), and CREB (Fig 4), respectively, and LPS/TLR4 signaling downregulated Slc22a17 ( Fig 5). Conversely, AVP reduced Lcn2 expression and predominantly apical Lcn2 secretion evoked by LPS, but through a posttranslational mode of action that was independent of cAMP signaling (Fig 6).

Comments : in the discussion, the authors are indicating “In support, a putative Nfat5 DNA consensus motif (osmotic response element (ORE) or tonicity-responsive enhancer (TonE)) [44] was identified in the human SLC22A17 promoter sequence ENSR00001455985/14:23820951-23822367 (828AGGAAAATGCCA839), which is compatible with the data obtained in Figures 3A and 3B. It would be worthwhile to test this putative ORE in future experiments by site-directed mutagenesis of the sequence in a Slc22a17luciferase-reporter gene assay.” Same interesting speculation and need to do luciferase-reporter experiments related to “a putative binding site at position -736” and “a putative NFAT5 DNA binding sequence in a region of the human Lcn2 gene (841TGGAAAAAGGCT852), which could explain Lcn2 downregulation by hyperosmolarity” Would you have any experiments here as well as any data on ko or conditional ko experiments related to Lcn2 or Lcn2-R?. A schematic representation of the new findings of the ms similar to Fig 7 of Betten et al 2018 could be helpful.

We have developed a nephron- collecting duct-specific ko mouse. Yet the problem with this ko mouse is that has decreased fertility, and so far we have not been able to generate a colony. The luciferase-reporter experiments are indeed useful and planned in the future studies. But we are unable to perform those experiments within reasonable time for revision of this manuscript.

Thank you. We have added a schematic representation of the data (Figure 7 of the revised manuscript), as suggested and mentioned it in the Conclusions (line 558 of the revised manuscript).